# Investigation of the Usability of Reduced Alkalinity Red Mud in the Building Material Industry

Miklós Jakab *, Gergely Balázs Patthy, Tamás Korim and Éva Makó

Department of Materials Engineering, Research Centre for Engineering Sciences, University of Pannonia,
Egyetem St. 10, H-8210 Veszprem, Hungary; patthy.gergely@gmail.com (G.B.P.);
korim.tamas@mk.uni-pannon.hu (T.K.); kristofne.mako.eva@mk.uni-pannon.hu (É.M.)
* Correspondence: jakab.miklos@mk.uni-pannon.hu

**Abstract:** Untreated and pH-reduced red mud is used as a potential raw material in ceramic technology. During the alkalinity reduction process, $CO_2$ is bubbled through the untreated red mud, which is particularly important as it can reduce the $CO_2$ content of the atmosphere, and the pH of the red mud. Therefore, this method serves as a $CO_2$ capture technique that utilizes waste as a raw material with low costs. Besides, reducing $CO_2$ emission, it allows the production of material suitable for brick manufacturing from waste. In this study, treated and reduced alkalinity red mud was mixed with clay in the range of 5–30 wt%, and the physical, chemical, mechanical, and technologically important properties of the dried and sintered bricks were examined. The application of reduced alkalinity red mud as an additive offers advantages, as the resulting bricks require less water for processing, are less sensitive to drying, and their strength values exceed those of the commercially available bricks. Therefore, the technique presented in the study enables the production of bricks and roof tiles with advantageous properties using waste materials.

**Keywords:** reduced alkalinity; red mud; clay; brick; waste recovery





## 1. Introduction

In recent decades, the aim of minimizing environmental damages, and the disposal and use of waste has gained prominence in the spirit of sustainable development. One of the most involved industries is building material production, with a significant amount of research already conducted regarding the production of bricks, roof tiles, concrete, and mortar from different types of waste [1].

From the beginning of using the Bayer process in aluminum production, a severe and in practice still unsolved problem has arisen: the treatment of the produced red mud [2]. This non-soluble waste poses a serious environmental problem, with 1–1.5 metric t of red mud produced for every metric ton of aluminum-oxide [3,4]. Currently, the most prominent and almost only way of disposal is depositing, which is still a cause of concern, due to the real and potential events of catastrophe, such as the leaking of the reserve, or the collapse of the holding dam.

One possible field of use for treating red mud is in the building industry, more specifically in brick production. The possibilities for using it in manufacturing technologies that are using sintered red mud by itself have been a subject of research since the 1980s [5–8]. Currently, the most effective field of use is as a mixing material in brick and roof tile production, where it can be mixed in large amounts with clay. Currently, there are two existing approaches in the industry, depending on whether the final product is fired or not. Traditional bricks are sintered at 1000 °C to reach the desired properties (strength, porosity, vapor permeability, color, etc.). For the production of non-fired products, an inorganic binding agent (such as cement) is used. Between 1972 and 2008, 14 patents were filed in the field of using red mud in the building ceramics industry, 3 for burned and 11 for non-burned products [9].

The already existing heterogeneity of natural clay grants the possibility of mixing in large amounts of red mud. The high iron, sodium, calcium, magnesium, and other melting point-reducing component content of red mud even makes it advantageous to mix in, particularly, if the final product is sintered [8,10].

These properties of red mud motivated researchers to conduct experiments on its use as a raw material. These experiments sought a way to use the red mud in itself and fully [5–9]. Xu et al. [9] sintered four differently composed red mud at 1140 °C. Their study showed that all products were suitable for internal and external brickwork. Moya et al. [5] reported that the optimal sintering temperature for bricks containing red mud should be at least 1200 °C, as this is the temperature where the optimal properties, such as the largest shrinkage and the lowest water absorption, can be achieved.

The work of Kavas [11] can be considered an in-between solution. While processing borate minerals, a significant amount of boron-containing matter remains as a byproduct. If this matter is mixed with red mud, advantageous building ceramic products can be obtained, because of the plasticizing effect of boron. Samples containing 15 wt% of red mud were fired at 900 °C and reached a bending strength of 45.3 MPa.

Another main solution is mixing red mud and clay together. According to several authors [12–18], this mixing step can improve the mechanical-technical properties of the product. Pérez-Villarejo [14] found that the mass fraction of red mud in clay should be 0.5 maximum for brick production. In this case, the optimal sintering temperature should be 950 °C, with one hour of holding time. The final products had a compressive strength of 52.5 MPa, comparable to bricks made of pure clay. He et al. [15] tried different mixing ratios, even a raw material with 80 wt% red mud in it. In this case, the samples were fired at 1050 °C for two hours. They found that the compressive strength of the bricks increased with the red mud concentration, but this increasing tendency stopped and reversed above 40 wt%. They explained that the compressive strength was affected by the amount of the glass phase being present. Ribero et al. [17] mixed 60 wt% red mud into clay, and they found that the maximal linear shrinkage, minimal water absorption, and maximal bending strength of 2.5 MPa was achieved by sintering the brick at 900 °C if 20 wt% of red mud has been added to the clay. Babisk et al. [16] wrote that the optimal mixing ratio for the clay-red mud system was 50 wt%. The largest measured pressing strength was around 13.1 MPa. The works listed above demonstrate that it is possible to use red mud as an additive if the red mud content is kept around 40–50 wt% and the firing is done at temperatures below 1050 °C [16].

Scribot et al. [18] used a modified red mud with reduced pH in their experiments. The production was done using the method developed by the company Alto. The aluminum oxide is extracted in the Bayer process with sodium hydroxide, which is reclaimed by the following method: the product is first washed with water, and then the residue is filtered under pressure. This process allows the production of red mud that contains less sodium hydroxide, which greatly helps in using it as an additive. They found that clay mixtures containing 30 wt% pH-reduced red mud sintered at 1015 °C showed excellent properties, with a maximal compressive strength of 64.9 MPa.

Apart from the article mentioned above, no other credible description has been given of a technology where the strong alkalinity of the used red mud is reduced, and this quasi-pH-neutral additive is mixed with the clay. pH is an important factor since the high pH of red mud makes large-scale production impossible due to the rapid and severe corrosion of the metallic structural equipment. Industrial experience has shown that the maximum usable red mud content is 5 wt%.

Although not directly related to bricks made by adding reduced alkalinity red mud, Inge Rörig-Dalgaard et al. [19] investigated the effects of different pH on bricks. During the experiments, the bricks were soaked in solutions with pH values of 3.5, 7, 9, 11, and 13 for 340 days. The acidic solutions were sulfuric acid, while the alkaline solutions were sodium hydroxide. In the pH = 13 environment, the concentrations of potassium, aluminum, and silicate ions increased significantly. The weight of the sample decreased by 2 wt%,

indicating that the potassium ions originated from the corrosion of the silicate phase. The physical properties did not change significantly compared to the bricks soaked in the neutral environment, except for the increased porosity and reduced bulk density to a lesser extent. Lewin and Charola [20] conducted similar experiments and found that with the increasing porosity and the increased reactivity of the decomposition products, water, and other soluble pollutants can penetrate the structure more easily, leading to further property degradation.

This study aimed to investigate new possibilities in the use of $CO_2$-treated red mud. It is important to reduce the pH of the raw materials as the production equipment is prone to corrosion. Moreover, the carbonates formed during the alkaline deoxidation process can positively impact the porosity of small solid bricks. The method presented here is a viable alternative to using $CO_2$ as a red mud semi-refiner, and could potentially utilize the large quantities of red mud waste available to capture the $CO_2$ emitted to the atmosphere. In terms of utilization, one of the most important parameters for small-sized solid bricks is compressive strength. Therefore, the mechanical properties of the test specimens prepared in the study were compared with the strength values of commercially available products in Hungary.

## 2. Materials and Methods

### 2.1. Materials

2.1.1. Western Transdanubian Clay from Devecser (C)

The natural clay used in the study is sourced from a brick factory operated by Leier Hungary Ltd. (Devecser, Hungary). It is located in the vicinity of the largest Hungarian alumina processing plant (Ajka, Hungary), as Hungary's biggest red mud reservoir (Kolontár, Hungary). Table 1 summarizes the phase composition of the clay, determined by the Rietveld method.

**Table 1.** The phase composition of the clay from Devecser.

| Phase | Chemical Formula | Weight% |
|---|---|---|
| calcite | $CaCO_3$ | 9 |
| albite | $NaAlSi_3O_8$ | 3 |
| kaolinite | $Al_4Si_4O_{10}(OH)_8$ | 2 |
| clinochlore | $Mg_5AlSi_3AlO_{10}(OH)_8$ | 6 |
| illite | $(K,(H_3O)^+)Al_2Si_3AlO_{10}(OH)_2$ | 16 |
| quartz | $SiO_2$ | 22 |
| dolomite | $CaMg(CO_3)_2$ | 8 |
| amorphous | - | 34 |

The main phase component of the clay is in amorphous form; however, it contains a large amount of quartz (JCPDS: 33-1161) and clay minerals such as kaolinite (JCPDS: 14-0164), clinochlore (JCPDS: 12-0243), and illite (JCPDS: 26-0911). In addition to clay minerals, the base material also contains a small amount of albite (JCPDS: 41-1480). Taking into account the amount of calcite (JCPDS: 5-0586) and dolomite (36-0426) minerals, the Devecser clay is clearly classified in the "calcareous clays" group. The analysis of the particle size distribution showed that the average diameter of the clay particles is $13.78 \pm 2.82$ μm, and the specific surface area is 17.33 cm$^2$/cm$^3$ (measured by laser diffraction).

2.1.2. Red Mud-Based Additives
Red Mud (RM)

One of the additives used in this study is red mud, which is a waste sludge produced during alumina production. The red mud used in the experiment was obtained from the tailings pond near the brick factory in Devecser (Kolontár, Hungary), and was utilized without further treatment. The red mud particles have an average size of $4.53 \pm 0.26$ μm, and the specific surface area is 41.12 cm$^2$/cm$^3$.

Reduced Alkalinity Red Mud (ARM)

The accumulated red mud over the years contains up to 40 wt% moisture, which is practically a solution of NaOH due to the production process. This results in a high pH value of freshly deposited red mud, which can be greater than 12. Over time, the moisture and alkalinity of the red mud decrease due to natural processes such as rainfall and reaction with $CO_2$ in the air, leading to the formation of $Na_2CO_3$. This process reduces the pH of the red mud to around 11, which is still too high for certain applications.

To neutralize the red mud, the NaOH content is converted to $Na_2CO_3 \cdot H_2O$ through the bubbling of carbon dioxide, into the red mud-containing tank, thus reducing its pH. Based on the work of Bonenfant et al. [21], in the production of ARM, 25 kg of wet red mud was used in a batch. During the alkalinity-reducing process, four, approximately 2 m long perforated steel pipes were placed in a tank containing red mud, and carbon dioxide gas was bubbled through them. Part of the resulting material was provided by MAL Ltd. (Hungarian Aluminum Production and Trading Ltd., Ajka, Hungary). This process can bring the pH down to around 8, which is less likely to cause corrosion. During this process, only the components of the red mud bound to $Na_2CO_3$ changed, while other components such as $Fe_2O_3$ remained practically unchanged. The reduced alkalinity red mud particles had an average size of $4.02 \pm 0.67$ μm, and the specific surface area was 39 $cm^2/cm^3$.

### 2.1.3. Morphology and Phase Composition of the Raw Materials

The secondary electron images representing the microstructure of the starting materials are summarized in Figure 1. The image of the clay clearly shows the layered structure characteristic of the phyllosilicates (Figure 1a). The untreated (Figure 1b) and reduced alkalinity (Figure 1c) particles of red mud had nearly identical sizes and shapes, consistent with the results obtained from laser particle size analysis.

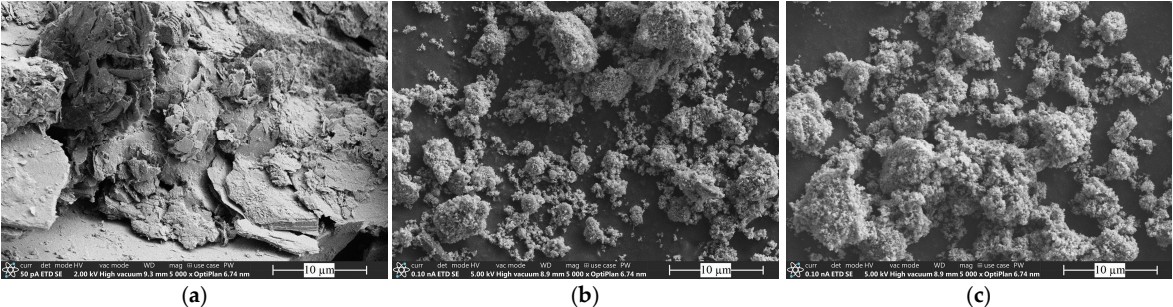

(**a**)         (**b**)         (**c**)

**Figure 1.** The scanning electron microscopic (SEM) images of the clay (**a**), RM (**b**), and ARM (**c**) samples.

The phase composition of the red mud additives is summarized in Table 2. The red mud additives comprised the following phases: hematite (JCPDS: 33-0664), cancrinite (JCPDS: 48-1862), forsterite (JCPDS: 34-0189), gibbsite (JCPDS: 33-0018), goethite (JCPDS: 29-0713), boehmite (JCPDS: 21-1307), rutile (JCPDS: 21-1276), calcite, and, due to the effect of treatment with carbon dioxide, the $CO_2$-treated red mud contained a thermonatrite phase (JCPDS: 8-0448). The composition of red mud can vary greatly and is determined by the type of bauxite. The bauxite found in Hungary mainly contains hematite, as well as smaller amounts of boehmite, forsterite, and rutile. The cancrinite phase present in the raw material origins from the processing of alumina. The calcite and gibbsite phases can originate from either the bauxite or the alumina production [22]. Based on the quantitative phase analysis determined by the Rietveld method, the additives contain mostly hematite and cancrinite, as well as more than 30 wt% of the amorphous phase. The reduced alkalinity red mud contains around 15 wt% of thermonatrite.

**Table 2.** The phase composition of the raw materials.

| Phase | Chemical Formula | Weight% | |
|---|---|---|---|
| | | RM | ARM |
| hematite | $Fe_2O_3$ | 33 | 31 |
| cancrinite | $Na_6Ca_2Al_6Si_6O_{24}(CO_3)_2$ | 12 | 10 |
| forsterite | $Mg_2SiO_4$ | 1 | 1 |
| gibbsite | $Al(OH)_3$ | <1 | 2 |
| goethite | $FeO(OH)$ | 8 | <1 |
| boehmite | $AlO(OH)$ | <1 | <1 |
| rutile | $TiO_2$ | 2 | 1 |
| calcite | $CaCO_3$ | 4 | 7 |
| thermonatrite | $Na_2CO_3 H_2O$ | - | 15 |
| amorphous | - | 39 | 31 |

The chemical composition of the raw materials is presented in Table 3. Previous studies have shown that if the $SiO_2/Al_2O_3$ ratio ratio is approximately 1.18, it indicates a high content of clay minerals in the raw material [23]; however, in the case of the applied Devecser clay, this ratio is 2.75. The $SiO_2$ content of the clay is quite high, which greatly influences the sintering processes that occur during heat treatment. Consistent with the X-ray diffraction phase analysis, the quantity of CaO and MgO in the clay represents the presence of dolomite and calcite. The $Fe_2O_3$ present in the clay is responsible for the red coloration that occurs during sintering and for modifying the glassy network structure.

**Table 3.** Chemical composition (wt%) and loss on ignition values (wt%) of the base materials.

| | Na₂O | MgO | Al₂O₃ | SiO₂ | SO₃ | K₂O | CaO | TiO₂ | Fe₂O₃ | LOI |
|---|---|---|---|---|---|---|---|---|---|---|
| **Clay** | 1.5 ± 0.3 | 5.9 ± 0.8 | 16.4 ± 1.2 | 45.2 ± 2.6 | 0.4 ± 0.1 | 2.9 ± 1.2 | 14.2 ± 0.7 | 0.7 ± 0.1 | 5.7 ± 0.9 | 6.69 |
| **RM** | 9.0 ± 0.8 | 1.3 ± 0.1 | 19.9 ± 2.2 | 11.0 ± 1.8 | 0.3 ± 0.1 | 0.2 ± 0.1 | 8.9 ± 0.9 | 3.7 ± 0.3 | 36.8 ± 2.5 | 8.77 |
| **ARM** | 11.0 ± 0.4 | 1.4 ± 0.2 | 17.6 ± 0.3 | 12.9 ± 0.3 | 0.5 ± 0.1 | 0.3 ± 0.2 | 8.2 ± 0.8 | 3.8 ± 0.1 | 33.1 ± 0.5 | 11.19 |

The quantities of main elements present in the red mud additives are almost the same. The quantity of oxides is determined by the quality of the bauxite used in the Bayer process. In the case of the $CO_2$-treated red mud, the quantity of $Na_2O$, and $Al_2O_3$ is almost identical. The network-modifying oxides ($Na_2O$, CaO, $Fe_2O_3$) present in the red mud can be used to influence the formation of glassy phases that occur during heat treatment by adding them to the clay. As a result of the heat treatment, a higher value of LOI (loss on ignition) can be observed in the reduced alkalinity red mud. This difference is presumably attributed to the thermal decomposition of thermonatrite. The amount of these oxides can therefore influence the firing temperature if the goal is to produce small, dense bricks.

### 2.1.4. Thermal Properties of the Clay and the Red Mud

In Figure 2, the thermoanalytical curves of clay and RM samples are shown. In the TG curve of the clay sample, the first mass loss (1.02%) occurred below 200 °C. This is due to the release of adsorbed and interlayer water. The second mass loss (1.77%) between 300 and 700 °C can be attributed to the dehydroxylation of clay minerals (illite, kaolinite, and clinochlore). The main mass loss (3.78%) occurred between 700–950 °C, which could be related to the decomposition of carbonates (calcite and dolomite). This decomposition caused a well-defined endothermic peak at 880 °C on the DTA curve of the clay sample [24]. The total ignition weight loss for the clay was 6.69%.

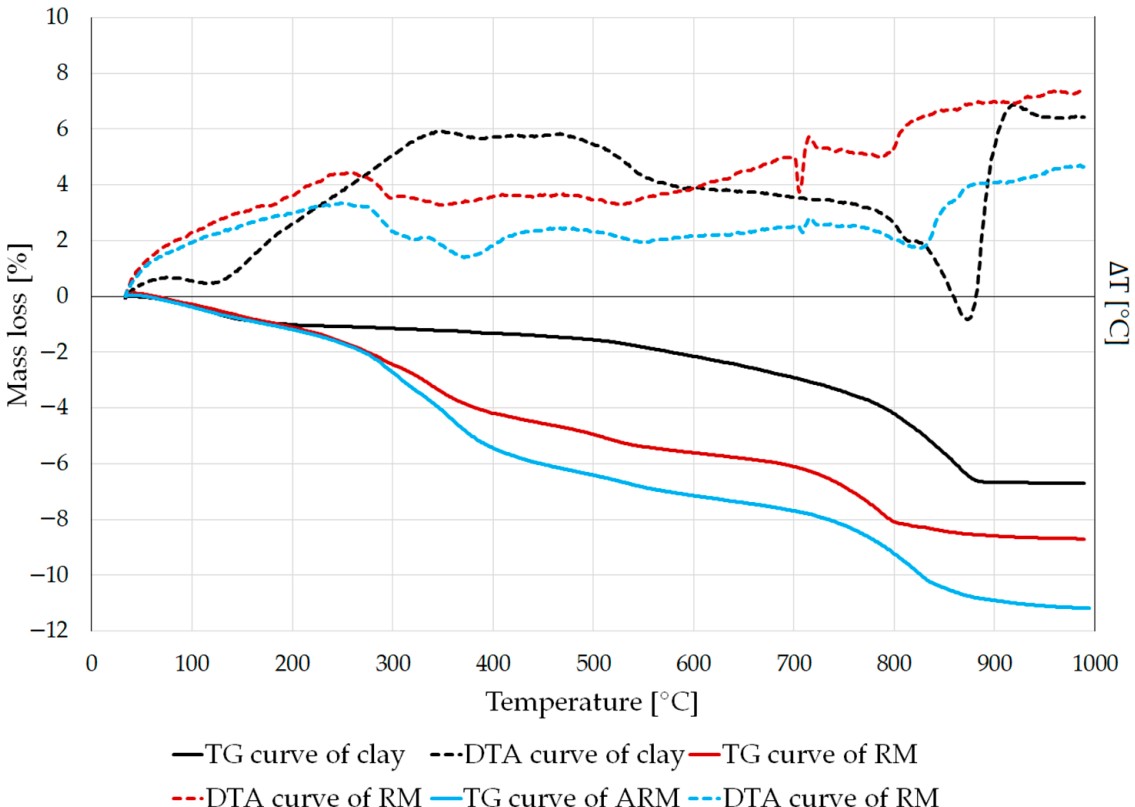

**Figure 2.** TG/DTA curves of the clay (indicated in black), the red mud (marked in red), and the reduced alkalinity red mud (marked in blue) samples.

In the TG curves of the RM and ARM samples (Figure 2), three mass loss steps can be determined. The first step below 500 °C may be related to the loss of adsorbed water and water molecules located in goethite, gibbsite, cancrinite, and thermonatrite. In this step, the mass loss of the ARM sample (5.13%) is higher than that of the RM sample (4.94%) because of the loss of water in the crystallization of thermonatrite. The second mass loss step between 500 and 650 °C corresponds to the decomposition of boehmite (0.87% for RM, and 1.19% for ARM). From 650 °C to 950 °C, the last mass loss (2.86% for RM, and 3.78% for ARM) could be attributed to the release of $CO_2$ from cancrinite, calcite, and thermonatrite [25,26]. For the untreated red mud, the total mass loss is 8.77%, and 11.19% for reduced alkalinity red mud.

*2.2. Methods*

2.2.1. Sample Preparation Method

Both starting materials were dried at 105 °C for 48 h and then ground, first in a jaw grinder and then in a cylindrical crusher, to produce particles smaller than 50 μm. The clay mixtures were then mixed with varying amounts (0%, 5%, 10%, 15%, and 30% by weight) of red mud additives, along with a specific amount of water to improve the workability of the mixture.

To determine the compressive strength and Pfefferkorn plasticity index, cylindrical test specimens with a diameter of 35 mm and a height of 40 mm were produced. Further tests were carried out on test specimens with dimensions of 80 mm × 40 mm × 20 mm. Generally, the typical size of small-sized solid bricks available in commercial trade is 250 mm × 120 mm × 65 mm, so the mechanical properties of the test specimens we produced can be compared with the values indicated in the manufacturer's datasheets.

After proper homogenization and molding of the raw materials, the specimens were dried at 105 °C to constant weight. The test specimens were then sintered at 850 °C, 950 °C, and 1050 °C, with a heating rate of 5 °C/min and 2 h of sintering time in an electric furnace

(Nabertherm Top 60, Nabertherm GmbH, Lilienthal, Germany). The composition of various red mud-containing clay mixtures is in given in Table 4.

**Table 4.** Composition of the different test specimens [wt%].

| Sample ID | Clay | RM | ARM |
|-----------|------|-----|-----|
| C | 100 | 0 | 0 |
| RM5 | 95 | 5 | 0 |
| RM10 | 90 | 10 | 0 |
| RM15 | 85 | 15 | 0 |
| RM30 | 70 | 30 | 0 |
| ARM5 | 95 | 0 | 5 |
| ARM10 | 90 | 0 | 10 |
| ARM15 | 85 | 0 | 15 |
| ARM30 | 70 | 0 | 30 |

### 2.2.2. Moisture Content

The wet weight of the samples was measured directly after molding ($m_w$). Then, the samples were dried at 105 °C until reaching constant weight and the dried weight was measured again ($m_d$). The workability moisture content of the samples containing different amounts of untreated and $CO_2$-treated red mud can be determined using the following formula:

$$moisture\ content\ (wt\%) = (m_w - m_d)/m_w \cdot 100. \tag{1}$$

### 2.2.3. Pfefferkorn Plasticity Index

The Pfefferkorn plasticity index is the moisture content given as a percentage by which a clay sample, with a size of ø35 × 40 mm, is compressed by a defined force (a dropping plate weighing 1190 g from a height of 146 mm) to a ratio of 1/0.307 of its original height. The plasticity index is related to the amount and quality of clay minerals in the sample. Different raw materials can be molded with varying amounts of water depending on their plasticity index.

### 2.2.4. Drying Sensitivity

In addition to shrinkage, an important property of clays is their sensitivity to drying, which refers to cracking and deformation during drying. The Macey method was used to characterize the drying sensitivity. For materials that are not sensitive to drying, the Macey drying sensitivity number is below 3.5 wt%.; for moderately sensitive materials it is between 3.6 and 7 wt%., for highly sensitive materials it is over 7.1 wt%., and for very highly sensitive materials it is over 10.1 wt% [27].

### 2.2.5. Linear Shrinkage

To determine the drying shrinkage of the test specimens, 50 mm long sections were created on each sample during the forming process and were measured again after drying and firing. The drying shrinkage of samples with different additive content can be calculated from the shortening of the section length during the drying process [28].

### 2.2.6. Apparent Porosity

The samples' apparent porosities and densities were measured using Archimedes' principle, a well-known method that uses water as the immersion liquid. The samples were sintered at various temperatures and spent 2 h in boiling water. The measurements were taken at room temperature.

### 2.2.7. Phase Composition

The untreated clay and sintered products were primarily investigated using X-ray diffraction (XRD) with a Philips PW 3710 powder diffractometer (PANanalytical, Almelo, Netherlands) equipped with a diffracted-beam graphite monochromator. The radiation used was CuKα (λ = 0.1541 nm). generated at 50 kV and 40 mA. A continuous scan mode was applied during the measurements, with a scanning speed of 0.02°2θ/s. Data collection was performed using X'Pert Data Collector software (version 2.0e). The HighScore Plus 5.0 software (Malvern Panalytical Ltd., Malvern, UK) was used to perform qualitative and quantitative phase analysis by the Rietveld method. The qualitative analysis of crystalline phases was executed by comparing the XRD patterns with the 2021 Powder Diffraction Files (PDF-2 2021) of the International Centre of Diffraction Data (ICDD).

### 2.2.8. Thermogravimetric Measurement

TG/DTA analyses were conducted using a Derivatograph Q 1500D-type instrument (Paulik-Paulik-Erdey, Nyíradony, Hungary) in a temperature range of 25–1000 °C, with high-grade corundum serving as a reference sample. In all cases, ~300 mg of the sample was heated in a corundum crucible under dynamic heating conditions (10 °C/min heating rate) and a static air atmosphere.

### 2.2.9. Particle Size Distribution

The particle size distribution was determined using a Fritsch Analysette 22 (Fritsch GmbH, Idar-Oberstein, Germany) laser particle sizer, which characterized the particles based on their intensity mean diameter.

### 2.2.10. Microstructure and Chemical Composition

SEM measurements were conducted using a ThermoFisher Apreo S (Thermo Fisher Scientific Brno s.r.o, Brno, Czech Republic) scanning electron microscope. The observations were made in high vacuum mode with an accelerating voltage of 20 kV. The chemical composition of the samples was analyzed using energy-dispersive X-ray spectroscopy (EDAX Ametek Octane Elect Plus, Ametek GmbH, Wiesbaden, Germany).

To determine the microstructure and porosity of the sintered products, a Nikon XT225 computer tomograph (Nikon Metrology Europe NV, Leuven, Belgium) was used. A beam current of 175 mA was used at an accelerating voltage of 225 kV for 3D reconstruction. Porosity analysis was carried out using Volume Graphics 3.4 software (Volume Graphics GmbH, Heidelberg, Germany).

### 2.2.11. Mechanical Properties

The compressive strength of the unfired and sintered cylindrical specimens, approximately 35 mm × 40 mm in size, was determined using an Instron 6800 type tester.

Three-point flexural strength tests were carried out using a TKI Plast-Bend-Tester testing machine on specimens sized 80 mm × 40 mm × 20 mm, with the load applied at the midpoint of the 80 mm × 40 mm surface. The flexural strength was calculated using the equation provided below:

$$\sigma = (3P_fL)/(2bh^2) \tag{2}$$

where $P_f$ is the load at fracture, $L$ = 80 mm is the sample length over which the load is applied, $b$ = 40 mm is the sample width and $h$ = 20 mm is the sample height.

### 2.2.12. AI-Assisted Technologies

The translation of the manuscript from the Hungarian language and the check for grammatical correctness were conducted using Chat GPT-3, an artificial language model developed by OpenAI.

## 3. Results and Discussion

### 3.1. Moisture Content

The moisture content values for each specimen are summarized in Figure 3. The water requirement for processing samples containing untreated and $CO_2$-treated red mud in varying amounts changes in a similar trend. In both cases, as the amount of red mud added to the clay increased, the amount of water required for processing decreased. This fact is advantageous for possible industrial use, since during the drying step following shaping, a smaller amount of water needs to be removed from the mass, which can save energy.

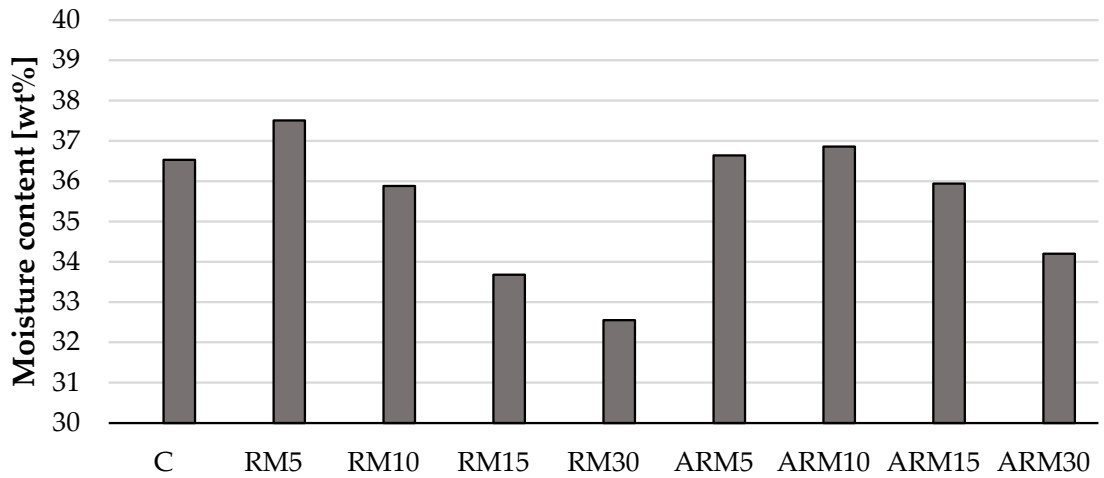

**Figure 3.** The amount of water required for processing the samples.

### 3.2. Pfefferkorn-Plasiticity Index

The results obtained from the determination of plasticity (Figure 4) showed that the Pfefferkorn plasticity number hardly changed at lower amounts of different red muds, and its values were practically the same. However, at 30 wt% amounts of untreated and reduced alkalinity red mud, its value increased slightly, indicating that adding red mud in such amounts increases the plasticity of the ceramic wet mass. Considering these facts, adding red mud is advantageous from a manufacturing technological aspect, as plasticity can be increased even further, thus making it possible to produce more complex, hollow products.

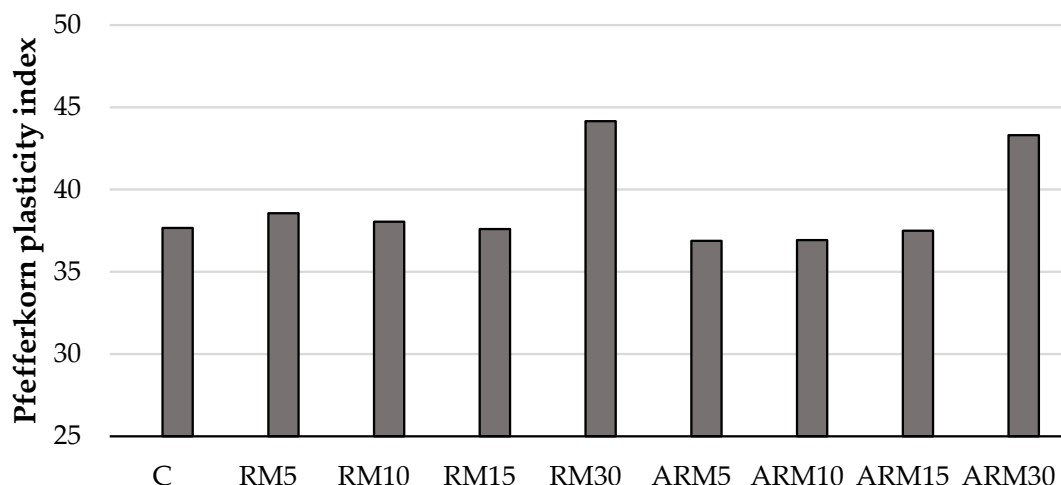

**Figure 4.** Pfefferkorn plasticity index for the different composition mixtures.

### 3.3. Drying Sensitivity

Figure 5 shows that all compositions were categorized as "not sensitive to drying". It can also be observed that increasing the amount of red mud reduces the Macey drying sensitivity numbers, which is advantageous from a technological perspective, as the test specimens are less prone to cracking and have a lower chance of developing microcracks and pores in their internal structure.

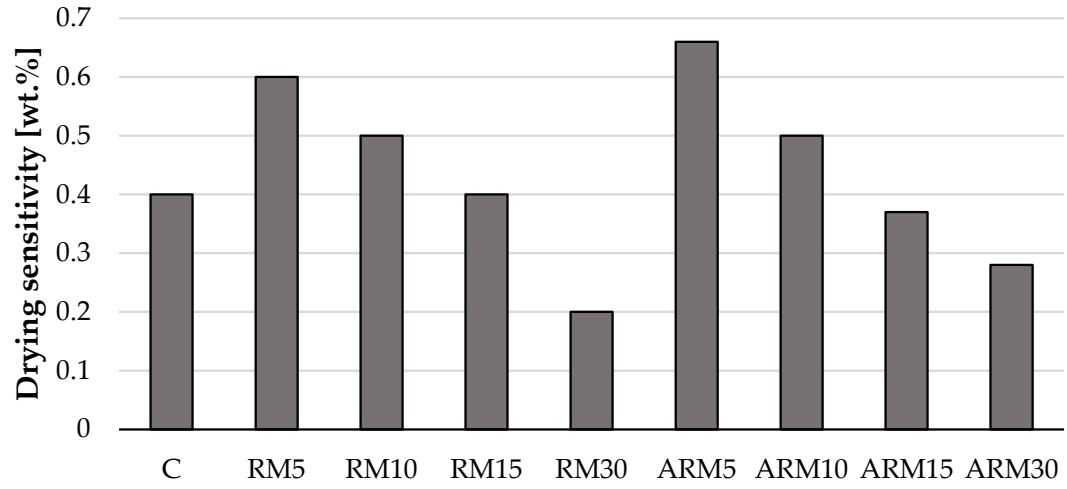

**Figure 5.** Drying sensitivity of the mixtures.

Previously, important technological parameters of the raw mixtures were presented. Overall, it can be said that adding red mud has a favorable effect on plasticity and drying sensitivity. Furthermore, in the case of bricks containing red mud, less water is needed for processing, which is important for reducing the energy required for drying while preserving the available stocks of water. There was no significant difference observed in the processing parameters between untreated and reduced alkalinity red mud during the investigation.

In the following sections, characteristic parameters of products sintered at different temperatures (850 °C, 950 °C, and 1050 °C) are presented.

### 3.4. Morphology and Phase Composition of the Sintered Bricks

Figure 6 shows the microstructure of the clay (C) sintered at a peak temperature of 1050 °C (Figure 6a), a brick containing 30 wt% untreated red mud (Figure 6b), and a specimen containing 30 wt% red mud treated with carbon dioxide (Figure 6c). The structure of the samples labeled RM30 and ARM30 indicates that the process of vitrification began at this firing temperature, resulting in a denser structure of the bricks. The amount of network-modifying oxides ($Na_2O$, $CaO$, $Fe_2O_3$) likely has a significant influence on the vitrification process compared to the raw clay. In the case of the untreated brick, many small pores were observed, while the red mud-containing samples had a denser structure. This phenomenon is expected to significantly affect the mechanical properties.

Table 5 shows the phase composition of the test samples fired at different temperatures. The main crystalline phases present in the samples are: diopside (D; JCPDS: 01-71-1067), feldspathic minerals (Fp: orthoclase; JCPDS: 31-0966, albite, anorthite; JCPDS: 41-1486), quartz (Q; JCPDS: 33-1161), mullite (M; JCPDS: 15-0776), gehlenite (Ge; JCPDS: 35-0755), hematite (H; JCPDS: 33-0664), nepheline (N; JCPDS: 35-0424), tridymite (Tr; JCPDS: 41-1401), muscovite (Mu; JCPDS: 7-0042), and amorphous phase (A). The results showed that as a result of the sintering quartz, gehlenite, and nepheline are likely to enter the molten phase, and at the firing temperature of 1050 °C, the mullite crystalline phase was formed. The firing process significantly increased the amorphous content of the samples, meaning that a glassy phase was formed, as confirmed by SEM images (Figure 6). When 30 wt% red mud-based additives were added to the clay, a significant amorphous phase was formed at

the firing temperature of 950 °C, and this phase content decreased slightly at 1050 °C. This is confirmed by the results of the Rietveld analysis.

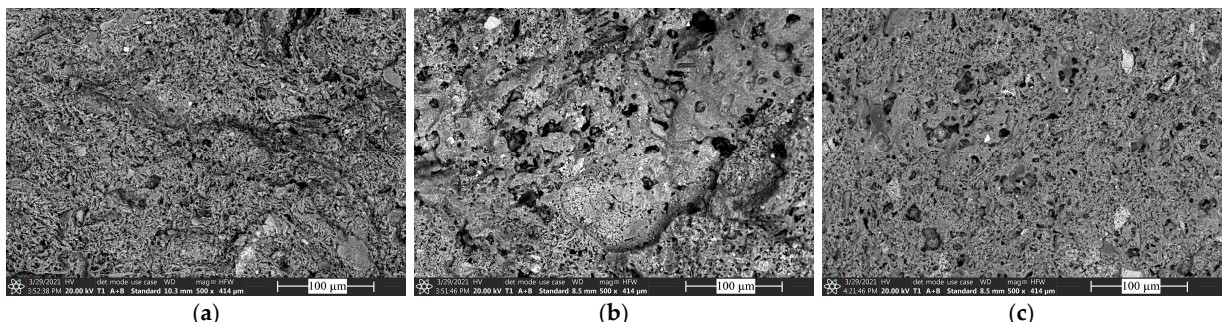

**Figure 6.** SEM images of the samples sintered at 1050 °C: C (**a**), RM30 (**b**), and ARM30 (**c**).

**Table 5.** The phase composition (wt%) of the sintered clay and the samples containing 30 wt% additives. The phase designations are as follows: D—diopside, Fp—feldspathic minerals (orthoclase, albite, anorthite), Q—quartz, M—mullite, Ge—gehlenite, H—hematite, N—nepheline, Tr—tridymite, Mu—muscovite, A—amorphous.

| Sample | D | Fp | Q | M | Ge | H | N | Tr | Mu | A |
|---|---|---|---|---|---|---|---|---|---|---|
| C-1050 | 31 | 33 | 8 | 2 | <1 | 1 | - | - | 1 | 26 |
| RM30-850 | 18 | 4 | 12 | - | 9 | 17 | 21 | - | <1 | 19 |
| RM30-950 | 19 | 7 | 5 | - | 3 | 11 | 14 | <1 | - | 41 |
| RM30-1050 | 30 | 19 | 2 | <1 | <1 | 10 | | <1 | - | 40 |
| ARM30-850 | 20 | 12 | 8 | - | 5 | 13 | 12 | - | - | 29 |
| ARM30-950 | 24 | 15 | 3 | - | <1 | 8 | 4 | <1 | <1 | 44 |
| ARM30-1050 | 24 | 31 | 1 | 2 | <1 | 10 | - | <1 | - | 31 |

### 3.5. Linear Shrinkage

In Figure 7, it is clearly visible that the increase in the content of red mud decreased the shrinkage of the test samples during firing at 850 °C and 950 °C compared to the additive-free clay. This is an advantageous property from a technological aspect, as it reduces the possibility of deformation and cracking. At 1050 °C, significant shrinkage was caused by the presence of large amounts of glassy (amorphous) phase (see Table 5). The diagram shows that the firing shrinkage was similar for the samples fired at 850 °C and 950 °C. For the samples fired at 1050 °C, the absolute value of shrinkage was greater, indicating densification, which is expected to result in greater compressive strength.

### 3.6. Porosity

The changes in the apparent porosity (Figure 8) can be divided into two parts. The data obtained at 850 and 950 °C indicate a property change opposite to the data acquired at 1050 °C. During firing at 850 °C and 950 °C, an increase in porosity was observed, which can be attributed to the release of $CO_2$ from the decomposition of carbonate minerals (dolomite, calcite, and thermonatrite). However, firing at the highest temperature led to the shrinkage of the specimens, induced by the increasingly developed molten phase. This is indicated by the increase in the amorphous fraction obtained from the quantitative evaluation of XRD data (Table 5), the evolution of firing shrinkage, and the morphology observed in the SEM images (Figure 6). The molten phase contracts the pores, partly filling them, explaining the increase in shrinkage, the decrease in porosity, and the subsequent changes in strength.

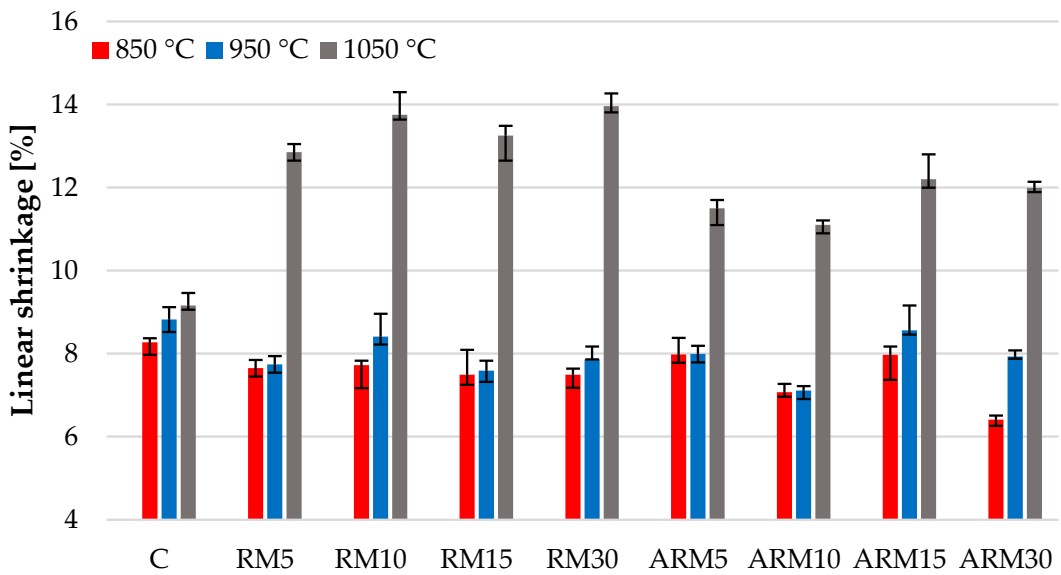

**Figure 7.** Linear shrinkage of the bricks sintered at different temperatures.

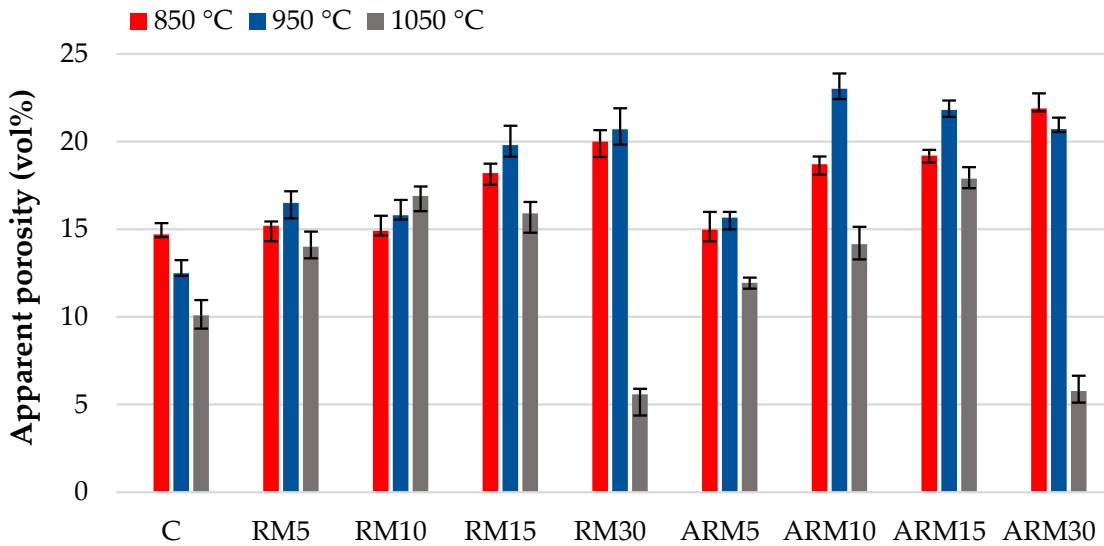

**Figure 8.** The apparent porosity of the bricks with different red mud content.

This fact can lead to two different directions for practical use, depending on the application: the production of a product with high strength or the manufacture of a building element with high porosity, providing good thermal insulation properties. Since achieving sufficient strength can be accomplished by using organic additives that burn out, creating porosity, it is more important to achieve the necessary strength than to create a large in-situ porosity. Based on this, sintering at 1050 °C seems to be the optimal choice, even with 30 wt% red mud content.

It is worth noting that red mud content exceeding 10 wt% represents a significant amount in terms of material volume, and there was a significant difference in the volume of clay and red mud added as an additive.

The results obtained from the CT scan are well correlated with the previous findings. The following Figures 9 and 10 shows the development of closed porosity in samples fired at different temperatures, using the ARM30 sample. The highest amount of closed pores is observed in the sample fired at 850 °C (Figure 10a), which decreased slightly in the sample fired at 950 °C (Figure 10b). At 1050 °C, the larger pores tended to merge, resulting in a more compact pore structure (Figure 10c). Although the apparent porosity

value decreased significantly, numerous small, closed pores were still present in the test samples, as observed in the scanning electron microscopy images (Figure 6).

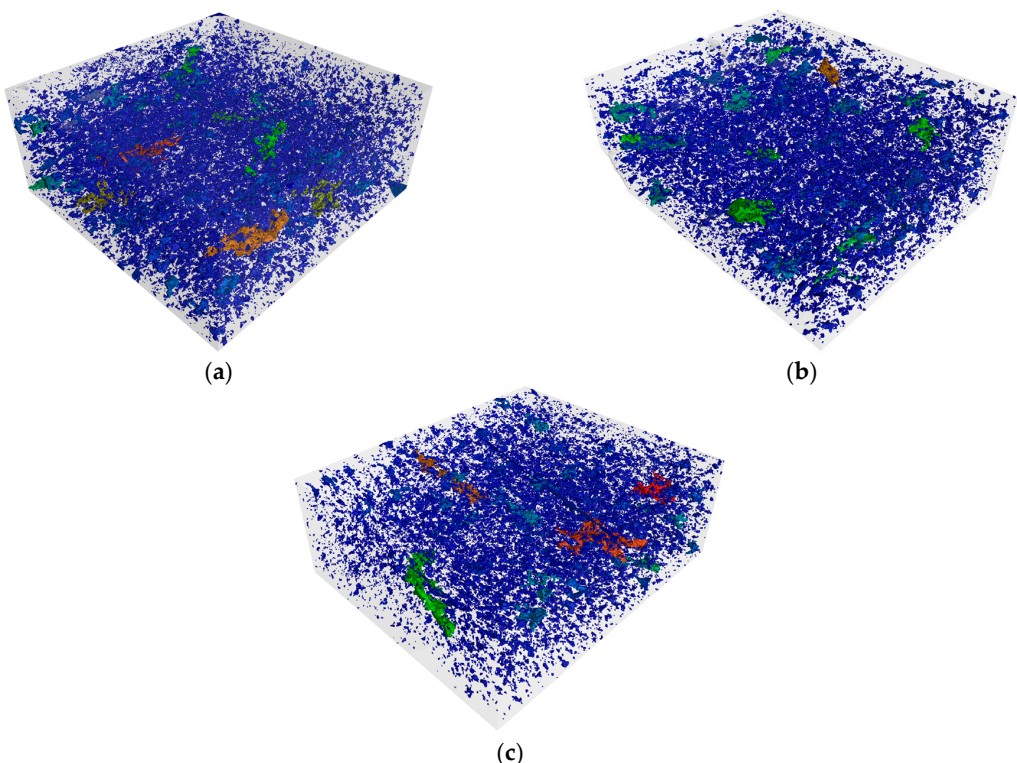

**Figure 9.** X-ray tomographic models of ARM30 samples sintered at 850 °C (**a**), 950 °C (**b**), and 1050 °C (**c**).

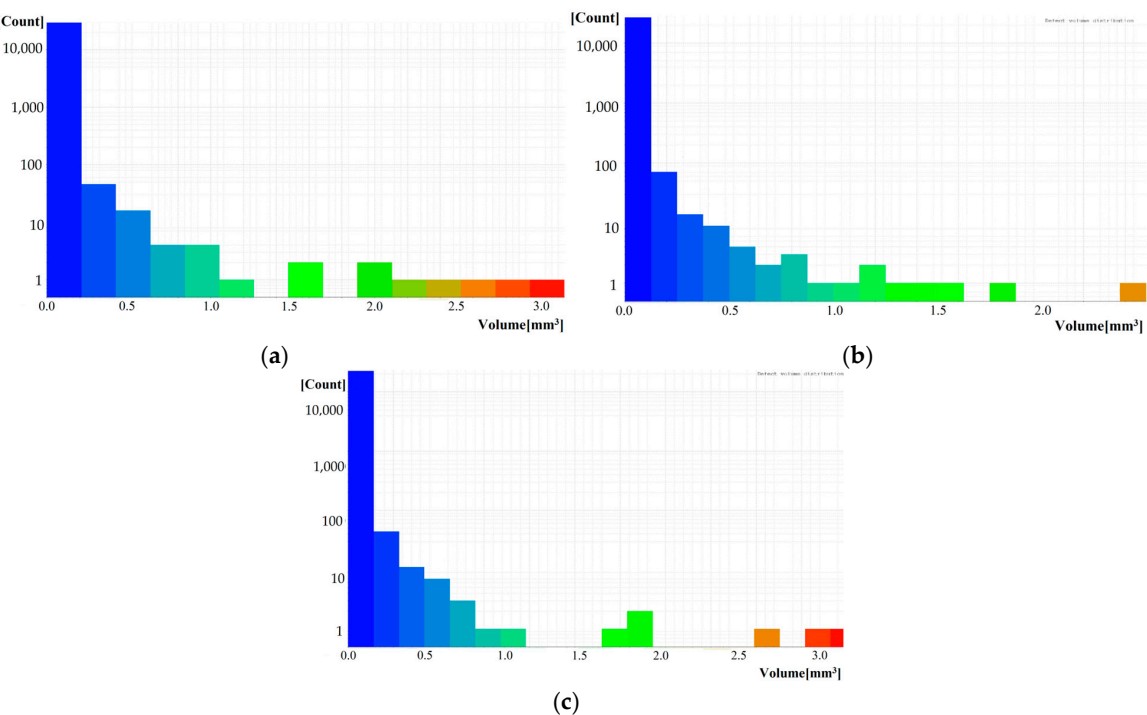

**Figure 10.** The pore size distribution curves of CT scans of ARM30 samples sintered at 850 °C (**a**), 950 °C (**b**), and 1050 °C (**c**) (pore size is represented on a color scale from blue to red).

During the closed porosity analysis carried out with X-ray tomography, blocks of 50 mm × 50 mm × 30 mm were cut using VG Studio 3.4 software. According to the closed porosity analysis performed on the blocks, the closed porosity of bricks fired at 850 °C was 2.04 vol%, while for the sintered brick at 950 °C, it was 1.77 vol%. In the case of the heat-treated brick at 1050 °C with a 30 wt% of alkalinity red mud content, the measured porosity was 1.66 vol%. For RM30 samples, the closed porosity values were 3.43 vol%, 3.05 vol%, and 2.13 vol% during sintering at 850 °C, 950 °C, and 1050 °C, respectively. As observed in the open porosity measurement, the proportion of closed pores decreased with higher sintering temperatures, which is consistent with the melting processes occurring during heat treatment.

### 3.7. Flexural and Compressive Strenght

Both flexural and compressive strength tests (Figures 11 and 12) showed similar trends, with the difference being that the strength-increasing effect of the compact structure formed at 1050 °C was more pronounced in the flexural strength. Figure 12 showing the change in flexural strength clearly indicates that the flexural strength of the clay samples with varying amounts of red mud content decreased by nearly half during firing at 850 °C and the bending strength of the samples fired at 950 °C also continuously decreased as the red mud content increased. The numerical values of the two samples fired at the lower temperatures were very similar, and considering the error margins in most cases, the values were practically identical. The highest strength values were measured in the case of bricks, which were sintered at 1050 °C. This fact can be explained by the strength-increasing effect of the amorphous/glassy phase that forms. However, it should be noted that according to X-ray diffraction analysis, the bricks fired at 950 °C (Table 5) contained the most amorphous phase, but these samples also showed the highest porosity values (Figure 8). The high degree of porosity is presumably due to the high carbonate content of the raw material (calcite, dolomite, and thermonatrite), whose decomposition has a pore-forming effect.

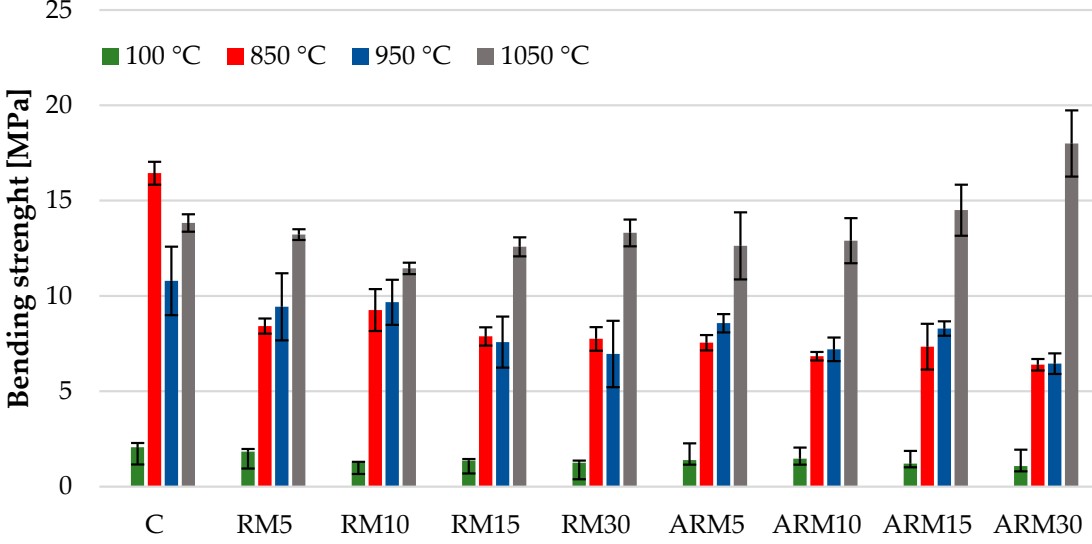

**Figure 11.** Flexural strength of the dried and sintered bricks.

These results correlate well with the changes in other physical and/or chemical characteristics previously presented, such as porosity, shrinkage, morphology, and composition.

In the case of compressive strength, the dominant strength-increasing effect of firing at 1050 °C is less clear, but even in this case, most of the samples had higher strength values. It is worth mentioning the reference strength value as well: the compressive strength of classic small-sized (250 mm × 120 mm × 65 mm) solid bricks rarely exceeds a value of 35 MPa (according to the datasheet for Leier Ltd. (L), Devecser, Hungary). For small-sized solid bricks distributed by Wienerberger Ltd. (W) in Hungary (Budapest, Hungary), the

standard compressive strength value is 32.4 MPa. Among other Hungarian brick factories, the manufacturer's datasheet for small-sized solid wall bricks distributed by Északmagyar Téglaipari Ltd. (ÉT) (Serényfalva, Hungary) indicates a compressive strength value of 20 MPa, while Pápateszéri Téglagyár Ltd. (P) (Pápateszér, Hungary) specifies a strength value of 18 MPa.

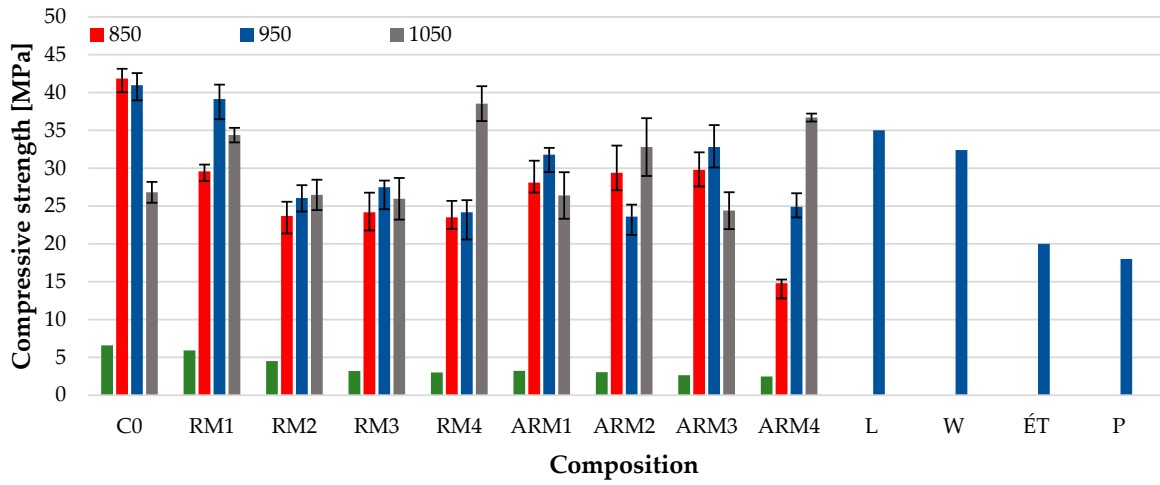

**Figure 12.** Compressive strength of the specimens.

For the bricks containing red mud, as presented in the study, the products fired at 950 °C and 1050 °C exceeded the strength requirements of the manufacturers described above. In this regard, the small-sized solid bricks produced by us may be used for outdoor masonry or decorative applications. In addition to manufacturing bricks with an appropriate strength, the addition of red mud is suitable for the production of ceramics with a deep red color for aesthetic purposes.

Therefore, it is clear that in cases where high alkalinity was neutralized in the samples, the remaining $Fe^{3+}$ and $Na^+$ ions were still capable of exerting their melt-forming effect, which is clearly evident at higher firing temperatures. Based on our research, it can be said that all of the red mud that undergoes the abovementioned treatment can be used in roof tile and brick production technologies. The reduced alkalinity method allows for significantly greater use of red mud (up to 30 wt%, instead of 5 wt%), as the relevant properties do not deteriorate and corrosive effects on steel structural materials are eliminated.

## 4. Conclusions

In this study, we presented the applicability of red mud treated with carbon dioxide in the production of building materials. The raw materials used in this study are readily available in significant quantities, with the Kolontár tailing pond and the Devecser brick factory located within 5 km of each other.

In summary, the following conclusions can be drawn from the results obtained during the research:

- Red mud can be advantageously used as an additive in the production of bricks since it significantly reduces the water demand of the manufacturing process, thereby reducing the energy required for drying;
- The reduced alkalinity red mud is particularly advantageous as it has a lower corrosive effect on the steel structures used in the manufacturing process, as well as being suitable for the production of more complex shapes based on the results of the plasticity tests and is less prone to cracking during drying;
- The porosity and strength of the sintered products can be fine-tuned based on the chosen additives and firing temperature according to the intended application;

- The compressive strength of the bricks produced, which contain red mud, can exceed the compressive strength of small-sized solid bricks available in the market, depending on the quantity of the additive used;
- The $CO_2$ treatment of the red mud also has great potential for developing a new inexpensive $CO_2$ capture method.

The material presented in this study is excellent for the production of small solid bricks and can also be used for roofing tiles if the product is fired to the proper density.

As each gram of red mud used reduces the consumption of the primary raw material, clay, the production costs of bricks decrease. This, in turn, leads to a decrease in mining and landscape destruction. Moreover, the utilization of red mud does not pose any foreseeable problems; for instance, it reduces storage costs. Therefore, the use of red mud is financially viable both directly and indirectly.

Taking into account factors such as the extraction and transportation costs of red mud, as well as other energy-intensive technological processes, a direct price reduction of 10–15% can be achieved with a 30% dosage. While this may not appear substantial, considering the massive quantity of bricks produced, the resulting savings can be significant. According to our estimates, around 30 million small-sized bricks can be manufactured in Hungary. If we calculate the price per piece at EUR 1, it amounts to approximately EUR 30 million. Even with a 10% price reduction taken into consideration, the savings would still be around EUR 3 million.

**Author Contributions:** Conceptualization. T.K. and M.J.; methodology. T.K.; validation. M.J. and G.B.P.; investigation. M.J. and G.B.P.; data curation. M.J.; writing—original draft preparation. M.J.; writing—review and editing. T.K., É.M. and G.B.P.; visualization. M.J.; supervision. T.K. and É.M. All authors have read and agreed to the published version of the manuscript.

**Funding:** This research was funded by GINOP-2.2.1-15-2017-00106 project.

**Data Availability Statement:** All data that support the findings of this study are included within the article.

**Acknowledgments:** In order to improve the English grammar, DeepL Translator and Chat GPT-3 were utilized. Chat GPT-3.0 specifically was employed to ensure linguistic accuracy.

**Conflicts of Interest:** The authors declare no conflict of interest.

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
