# Peer review of "Investigation of the Usability of Reduced Alkalinity Red Mud in the Building Material Industry"

_resources, doi:10.3390/resources12070079_

Round 1

Reviewer 1 Report

The paper aims to investigate the usability of alkali-reduced red mud in the building material. The paper is well organized and the results are interesting. However, the paper should be revised before it desired to be published. The detailed suggestions are listed as follows.

1 In table 3, why do the chemical components of mud change so greatly after carbonation? Especially Na2O and Al2O3.

2 Section 2.2.4, the reference to the Macey method for drying sensitivity should be given.

3 Section 2.2.5, the linear shrinkage test method should be described in more detail or given the related reference.

4 Section 2.2.6, the apparent porosity test method should be described in more detail, such as its immersion time.

Reviewer 2 Report

This study investigated the usability of CO2 treated red mud as construction material. This work is interesting and might be considered for publication after some concerns can be addressed.

1. Abstract must be revised. The abstract should start with a clear and concise statement of your research problem or objective, then summarize your methods briefly, and include any significant or novel findings and their implications for your research problem. Finally conclude with a statement about the implications of your research.

2. Authors indicate that the red mud was CO2 treated, which is also considered as the novelty of this work. However, I did not find any description on CO2 treatment in the manuscript. Authors may include a separation section on CO2 treatment.

3. What was the purpose of using heat-treated specimens?

4. CT scan esults need to be better presented. Current 3D image can barely show the difference of pore structures in different samples.

5. Conclusion should be presented as bullet point as it will make conclusion clearer.

6. Some reference might be helpful to improve the manuscrip quality:

 10.1016/j.matdes.2018.06.045

10.1061/(ASCE)MT.1943-5533.0003288

10.1061/(ASCE)MT.1943-5533.0002616

10.1061/(ASCE)MT.1943-5533.0004414

Language might be improved.
